

# On the evaluation of the phase relation between temperature and wind tides based on ground-based measurements and reanalysis data in the middle atmosphere

Kathrin Baumgarten[1] and Gunter Stober[1]

[1]Leibniz-Institute of Atmospheric Physics at the University of Rostock, Kühlungsborn, Germany

**Correspondence:** Kathrin Baumgarten (k.baumgarten@iap-kborn.de)

**Abstract.** The variability of the middle atmosphere is driven by a variety of waves covering different spatial and temporal scales. We are diagnosing the variability of the thermal tides due to changes in the background wind by an adaptive spectral filter, which takes the intermittency of tides into account. We apply this diagnostic to temperature observations from a daylight-capable lidar at mid-latitudes (54° N, 12° E) as well as to reanalysis data of horizontal winds from MERRA-2. These reanalysis
data provide additional wind information in the altitude range between 30 and 70 km at the location of the lidar as well as on a global perspective. Using the global data gives information of the tidal modes seen at one location. A comparison of the temperature and wind information affirms whether there is a fixed phase relation of the tidal waves in the temperature and the wind data. We found that in general the local tidal signatures are dominated by migrating tidal modes and the signature is weaker in temperatures than in winds. While the meridional wind tide is leading the zonal wind tide by 90°, the phase relation
between the temperature and the wind tide is more complex. At certain altitudes the temperature tide follows the zonal wind tide. This knowledge helps to improve the interpretation of the seasonal variation of tides from different observables especially when only data from single locations are used. The findings provide additional information about the phase stability of tidal waves and the results clearly show the importance of a measurement acquisition on a routine basis with high temporal and spatial resolution.

## 1 Introduction

Atmospheric waves play a major role in the circulation of the Earth's atmosphere as they couple the different atmospheric layers. These waves are often generated in the troposphere and propagate to higher altitudes, where they become dynamically unstable, break and deposit their momentum and energy to the mean flow (e.g., Fritts and Alexander, 2003). Especially such waves driven by the absorption of solar radiation are one of the strongest perturbations in the middle atmosphere. These so-
20 called thermal tides are global-scale waves with periods of one solar day and the subharmonics of it (24 h, 12 h and 8 h) (e.g., Chapman and Lindzen, 1970; Forbes, 1984; Hagan and Forbes, 2002). If tides propagate synchronously to the Sun (westward) around the globe they are usually called migrating, while tides are called non-migrating when the horizontal propagation is in the opposite direction (eastward) or stationary. Tides which also propagate westward but with a zonal wave number other than one for the diurnal or other than two for the semidiurnal component are also called non-migrating. Such tides are assumed to be





excited mainly by zonally asymmetric variations in topography or time varying heat sources (Lieberman et al., 2015; Sakazaki et al., 2015).

Global tidal fields have recently been extracted from satellite instruments. For example, tidal climatologies in the mesosphere and lower thermosphere (MLT) region have been constructed by temperature/wind observations from the High Resolution Doppler Interferometer (HRDI), the Wind Imaging Interferometer (WINDII) and the Microwave Limb Sounder (MLS) on board of the UARS satellite, or from the TIMED Doppler Interferometer (TIDI) and from the Sounding of the Atmosphere using Broadband Emission Radiometry (SABER) instrument on board of the TIMED satellite (e.g., McLandress et al., 1996; Forbes and Wu, 2006; Sakazaki et al., 2012; Pancheva et al., 2013). Most satellite studies have focused on the MLT region due to the large effect of tides at these altitudes. Only few satellite studies deal with the lower atmospheric region (Zhang et al., 2006; Mukhtarov et al., 2009; Sakazaki et al., 2018). However, satellites typically need several weeks to cover 24 h of local time at one particular location. Therefore, tidal results from satellite measurements are typically based on monthly mean values. Consequently, the short-term variability on timescales less than a month cannot be resolved from those instruments. Nowadays, there are a few approaches to extract the short-term variability of tides from satellite data using a deconvolution method for non-migrating tidal modes or with the combination of ground-based radar and satellite observations (Oberheide et al., 2002; Lieberman et al., 2015; Pedatella et al., 2016; Zhou et al., 2018). But these approaches are limited to lower latitudes ($<50°$ N) resolving only non-migrating tides or they are limited to the MLT region (Oberheide et al., 2002; Zhou et al., 2018). Therefore, these methods are not suitable to resolve the day-to-day variability of tides at latitudes of $54°$ N over the whole altitude range of the middle atmosphere.

In contrast to satellite data, ground-based observations have the advantage of a high temporal and spatial resolution in deriving tidal information at a certain location. Tidal observations require a full day data coverage, which is achieved using daylight-capable lidars or radars (Fong et al., 2014; Kopp et al., 2015; Jacobi, 2012; Pokhotelov et al., 2018). Radars typically provide quasi-continuous data sets of horizontal winds, but unfortunately these instruments cover only a limited altitude range between approximately 70 and 110 km in the MLT region (Hoffmann et al., 2010; Wilhelm et al., 2017). A lidar using Rayleigh scattering is also able to provide temperature data in the altitude range below the mesopause whenever the weather is appropriate, and thus, also the stratosphere and mesosphere is covered. Based on multi-day time series of temperature data a lidar provides useful case studies for investigating the short-term variability of tides. But solely from temperatures this will resolve only one part of the wave activity because the wind information is missing. A combination of temperature and wind information would yield a deeper understanding of the dynamics including wave dissipation and wave filtering mechanisms as well as the phase relation between the different observables. Therefore, this study combines a ground-based lidar with re-analysis data. Reanalysis data provide additional data where observations are missing by assimilating existing observations combined with an underlying forecast model resulting in global gridded data sets for a broad range of variables. For instance the Modern-Era Retrospective Analysis for Research and Applications version 2 (MERRA-2) produces temperature and wind fields over several years from the surface up to the lower mesosphere (Gelaro et al., 2017). Sakazaki et al. (2012, 2018) have investigated the representation of tides in several reanalysis data sets as well as in satellite data. In general, the results of the studies show a consistent diurnal migrating, semidiurnal migrating as well as a non-migrating tidal representation in all data





sets used from a global perspective. But there are differences in amplitude and phase for the diurnal migrating tide between reanalyses and SABER due to a trapped tidal mode.

The dynamic fields of temperature and wind fluctuations induced by tides are similarly related to the polarization relation of gravity waves. Such a polarization relation for tides has been theoretically derived from the primitive equations for fluctuations

of a dissipation-less atmosphere by She et al. (2016). They have shown that there is a fixed relation between both wind components and the temperature fluctuations induced by tides from the investigation of Na lidar measurements and model predictions from the Climatological Tidal Model of the Thermosphere (CTMT) based on the Hough Mode Extension (HME) technique in the MLT region. A derivation and affirmation of a tidal polarization relation would lead to a possibility to deduce an unknown tidal field from another observed tidal field and is therefore highly recommended also for the altitude range below

the mesopause region. For a westward propagating monochromatic wave with a single zonal wave number $s$ the polarization relation between the temperature tide $\widetilde{T}$ and the zonal wind tide $\widetilde{u}$ derived by She et al. (2016) is

$$\widetilde{T} = \frac{H_s}{R}\left(\mathrm{i}\widetilde{m}(z) + \frac{1}{2H}\right)\frac{\sigma^2 - f^2}{\sigma k}\widetilde{u} \tag{1}$$

with the horizontal wave vector $k = \frac{2\pi s}{2\pi a \cos\phi}$, the vertical wave number $m$, the Earth radius $a$, the latitude $\phi$, the Coriolis frequency $f$, the angular frequency of the tidal wave $\sigma$, and the scale height $H_s$ at a reference altitude. This relation is not valid

for a so-called trapped tidal mode as well as non-migrating tides. Consequently, it has to be proven if the relation still holds for tidal observations in the stratosphere and lower mesosphere, where those tides can occur (e.g., Forbes and Garrett, 1979).

Therefore, this paper presents tidal signatures derived from temperature and wind data using a Rayleigh-Mie-Raman (RMR) lidar and reanalysis data from MERRA-2 covering an altitude range between 30 and 70 km during a multi-day time series of observations in May 2016 at 54° N, 12° E. To our knowledge, this 230 h long data set is still the longest continuous data

set retrieved by a RMR lidar. The daylight capability of this RMR lidar as well as exceptionally good weather conditions made it possible to investigate wave structures over this time period, which allowed to study not only the short-term variability of gravity waves and tides, but also the tidal phase progression. The short-term variability of the atmospheric waves during this case study in May 2016 has already been described in Baumgarten et al. (2018) showing a large variability also for tidal signatures which coincides with an increase of gravity waves exactly when and where the diurnal tidal component showed

a strong decrease. However, the current paper focuses on the tidal phase relations observed in different quantities. A unified approach is presented in this study to retrieve tidal fields of different data sets from different instruments. This approach is able to deal with data gaps and unequally sampled data, which is often the case for observational data, and it enables a decomposition of the data into a tidal field, a background due to planetary waves as well as due to gravity waves with resolving amplitudes and phases during the sounding period.

The organization of the paper is as follows. In Section 2 we describe the different data sets for temperature and wind information using measurements as well as reanalysis data and how the data are treated using a unified diagnosing tool for tides. Section 3 presents the results for the tidal analysis as comparison of the tidal fluctuations simultaneously seen in temperature and wind data during the case study in May 2016. The results of the phase relation are discussed in Section 4. Finally, the findings are summarized and a conclusion is given in Section 5.





## 2 Description of the data and analysis methods

### 2.1 Data

The temperature data used in this study were obtained by the RMR lidar at Kühlungsborn (54° N, 12° E). This lidar was developed in 2009/2010 to measure the backscattered signal independently from any daylight conditions. Therefore, a commercial
frequency-doubled Nd:YAG laser at 532 nm is used as emitter. The laser beam is guided co-axially with the receiving telescope into the atmosphere. The field of view (FOV) of the receiver is limited by a fiber cable with a small core diameter of 0.2 mm, resulting in a small field of view of only 62 $\mu$rad. The advantage is a reduction of the scattered background light from the Sun. A narrow band interference filter (IF) as well as a double Fabry-Pérot-etalon (FPE) is used for spectral filtering. The IF has a full-width-at-half-maximum (FWHM) of about 130 pm, the double etalon of about 4 pm, respectively. A detailed description
of the lidar has been given in Gerding et al. (2016).

From the range corrected backscattered signal absolute temperatures are derived assuming hydrostatic equilibrium (Hauchecorne and Chanin, 1980). The initial temperature value for integration is taken from CIRA-86 (Fleming et al., 1990) in an altitude range between 70 and 75 km for the whole day due to the strong solar background at the highest solar elevation. The temperatures become independent from the start temperature approximately 1-2 scale heights below the initial retrieval altitude.
The integration time to retrieve the temperatures is 2 h with a temporal shift of 15 min. The vertical resolution is 1 km. Due to additional aerosol scattering below 30 km only lidar temperatures above this altitude are taken into account in this paper.

NASA's Global Modeling and Assimilation Office (GMAO) has produced the Modern-Era Retrospective Analysis for Research and Applications version 2 (MERRA-2), which assimilates a variety of observations from radiosondes, satellites, ships and also from land observations (Gelaro et al., 2017). The improvements of MERRA-2 compared to the former version are
the assimilation of aerosol observations and better representations of the stratospheric ozone and cryospheric processes. In addition to this, some jumps, trends and biases in aspects of the water cycle in MERRA-2 are reduced to improve the quality of the reanalysis data. The underlying general circulation model is the Goddard Earth Observing System (GEOS) version 5, which was improved in the dynamical core and the physical parameterization scheme. With the assimilated observational data the model provides, including the first version of MERRA, time series of data products from 1979 to present with a horizontal
resolution of 0.5° × 0.625°. The data products have 72 model and pressure levels from the surface to 0.01 hPa (0.1 - 75 km) with a variable vertical resolution between 100 m in the troposphere and up to 4 km in the lower mesosphere. The time series of temperature and wind data from MERRA-2 used for this study has a temporal resolution of 3 h.

### 2.2 Methods for the tidal analysis

Time series of temperatures and winds observed by ground-based instruments reveal dynamical processes in the atmosphere
over a certain altitude range. Those data sets contain an observation of a superposition of a slowly varying background which is disturbed by small scale waves like gravity waves. To decompose such temperature and wind data sets some a priori knowledge is necessary to retrieve tidal signatures. By assuming fixed periods as well as assuming large vertical wavelengths for thermal tides, we have developed an adaptive spectral filtering technique (ASF) to decompose the time series into a background (for





temperature $T_0$, zonal wind $u_0$, meridional wind $v_0$) and the tidal components ($T_{\text{fil}}$, $u_{\text{fil}}$, $v_{\text{fil}}$) according to the following equation

$$T_{\text{fil}}, u_{\text{fil}}, v_{\text{fil}} = T_0, u_0, v_0 + \sum_{n=1}^{3} \left( A_n \cdot \sin\left(\frac{2\pi}{P_n} \cdot t\right) + B_n \cdot \cos\left(\frac{2\pi}{P_n} \cdot t\right) \right) \tag{2}$$

where $P_n = 24, 12, 8\,\text{h}$ and $A_n$ as well as $B_n$ denote the coefficients for the tidal amplitudes based on temperatures, zonal or

meridional winds. A detailed explanation is given in Stober et al. (2017) but without taking into account the information about the vertical wavelengths. In a first step, we fit all coefficients to derive mean temperatures and winds (zonal and meridional component) as well as tidal amplitudes and phases for the diurnal, semidiurnal and terdiurnal component. The initial window length is 2 days. In further steps, the window size is adapted to the number of wave cycles to better account for the intermittency of waves. Therefore, the second step uses a window length of 24 h and the semidiurnal tide and terdiurnal tidal amplitude and

phase are fitted using the mean and diurnal tidal amplitude and phase as regularization constraint. The terdiurnal tidal amplitude and phase is obtained with a window length of 16 h inserting now the mean, the diurnal and semidiurnal amplitude and phase as regularization constraint. This procedure is repeated for every data point of the time series. Thus, there is a resulting time series of mean temperatures and winds, diurnal, semidiurnal and terdiurnal amplitudes and phases with the same resolution as the observations. The remaining variety is assumed to be due to gravity waves.

It turns out, that the adaptive spectral filter, if applied just in the time domain without further knowledge of vertical wavelengths does indeed provide a proper decomposition of the mean winds and the tidal components. But, it appears that gravity waves with similar periods like tides (especially with a periods of $\approx 12\,\text{h}$), but with smaller vertical wavelengths are interpreted also as tides. To avoid this, we introduced a vertical regularization in the fitting procedure assuming that the mean temperature and wind only show a small change with altitude. Further, we assume that the vertical wavelength of tides is larger than that of

gravity waves with periods around 12 h. We use a 8 km vertical regularization for the mean state and a 16 km vertical regularization for the tidal phases. These thresholds are based on the wavelet analysis presented in Baumgarten et al. (2017). In this study it has been shown that such inertia gravity waves have typically vertical wavelengths of about 8 km, while gravity waves with much smaller periods have comparably large vertical wavelengths in the range of those from tides. Studies by Mukhtarov et al. (2009) and Kopp et al. (2015) as well as model results from Forbes (1984) have revealed vertical wavelengths of the

diurnal tide between 15 and 50 km or even larger wavelengths for the semidiurnal tide. The vertical wavelength is in general largest for the first Hough modes and decreases with higher modes (Forbes, 1995). But observations suggest that only the first Hough modes are relevant in the atmosphere. Therefore, it is a suitable assumption to use 16 km as vertical regularization constraint. The regularization is implemented by allowing only a gradual change of the mean amplitudes or tidal phases for each vertical cut off wavelength, respectively. It appears that this approach is more suitable to separate also gravity waves

with shorter vertical wavelengths from tides. The advantage of this method is the uniform applicability for different data sets compared to a single vertical filtering or a separate filtering in the vertical and in the time as usually done for lidar data (e.g., Ehard et al., 2015; Baumgarten et al., 2017). Another advantage compared to Fourier based methods is that the technique is applicable to unevenly sampled data and data gaps. Further, it is possible to conduct an error propagation of the statistical uncertainties associated to each measurement of the time series.



On a global scale we use a least squares fit to extract the individual tidal components provided by MERRA-2 according to separate between migrating and non-migrating thermal tides and to classify the overall tidal structure in the atmosphere. Therefore, equation (2) is modified in the following way

$$T_{\mathrm{fil}}, u_{\mathrm{fil}}, v_{\mathrm{fil}} = T_0, u_0, v_0 + \sum_{s=-3}^{3} \sum_{n=1}^{2} \left( A_{sn} \cdot \sin\left( s \cdot \lambda - \frac{2\pi}{P_n} \cdot t \right) + B_{sn} \cdot \cos\left( s \cdot \lambda - \frac{2\pi}{P_n} \cdot t \right) \right) \tag{3}$$

where $s$ and $\lambda$ denote the zonal wave number and the longitude, respectively.

## 3 Results

In the first part of the results, the total tidal fields containing the diurnal, semidiurnal and terdiurnal component from temperature observations of the lidar will be compared to the tidal fields from temperature and winds obtained by MERRA-2 using the same analysis approach. The second part will be focused on the amplitudes and phases of the different tidal components based

on the mean values over the sounding period as well as time resolved. In the third part, the global fields from MERRA-2 will be investigated with a similar approach to determine which tidal components are seen in the local data.

### 3.1 Tidal fluctuations

Based on the filtering with the ASF the resulting tidal temperature field compared to the total temperature field from the lidar observation is shown in Fig. 1 a and b over a sounding period of approximately 10 days in May 2016 in an altitude range

from 30 to 70 km. The large scale structures from the harmonic analysis reveal a decay between 10 and 11 May, which was already investigated in Baumgarten et al. (2018) using only a 1-dimensional spectral filter. But the adaptive spectral filtering leads to more information about the amplitudes and phases in time and space, what is an enhancement compared to using the 1-dimensional filtering approach. In addition to the general behavior of the tides, it is clearly seen that the dominating variability of the temperatures is caused by tides. Especially, huge tidal amplitudes are visible in the stratopause region (around 50 km)

with a very stable phase during the sounding. Before a comparison of the temperature with additional wind information from MERRA-2 is done, we show, that MERRA-2 data in general captures the same tidal structures as the lidar. Therefore, also the MERRA-2 temperatures are analyzed using the adaptive spectral filter combined with vertical filtering. The temperature field as well as the filtered temperature field (tides and background) from MERRA-2 are shown in Fig. 1 c, d. From the absolute temperatures it can clearly be seen that in principle the resolved range of temperatures is almost the same in both data sets. But

MERRA-2 temperatures are approximately 5 K smaller than the lidar temperatures over the whole time (see Fig. 1 a, b at 40 km altitude). The filtered temperature fields show the same. The underestimated background temperature in MERRA-2 does not affect the temporal evolution of the tidal structures themselves. The tidal structures from the lidar and from MERRA-2 show a good agreement regarding the phase. Some small scale structures seen in the lidar data are not fully captured by MERRA-2 due to the coarser resolution of MERRA-2 data compared to the lidar data.

Subtracting the mean background state of the atmosphere from the filtered tidal temperature fields yield the temperature fluctuations only induced by tides. These fluctuations are shown in Fig. 2 for both data sets. Basically, there is a very good





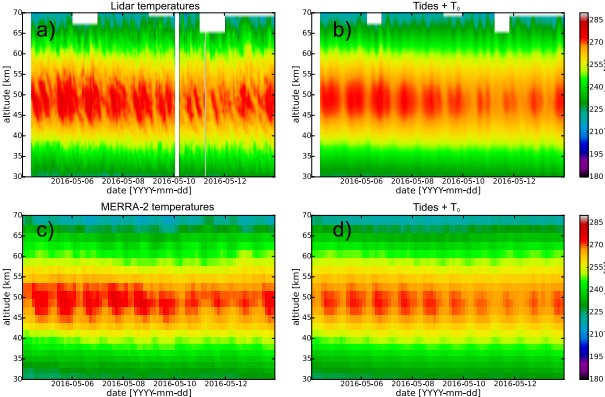

**Figure 1.** Temperatures and filtered temperature field including tides and background observed by the RMR lidar at Kühlungsborn on 4–13 May 2016 (a, b) and the corresponding data obtained by MERRA-2 (c, d).

agreement between both tidal temperature fluctuations. Because of the slightly better temporal resolution and the better vertical resolution of the lidar data compared to the MERRA-2 data, differences occur mainly for the subharmonic tidal components. Especially at altitudes larger than 60 km further discrepancies are visible due to the increasing amplitudes in the lidar data,

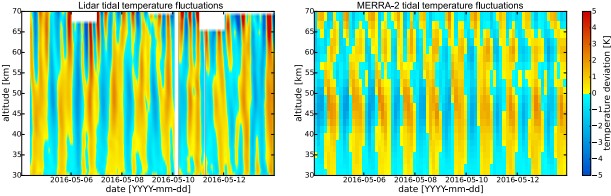

**Figure 2.** Tidal temperature fluctuations from the lidar (left) and from MERRA-2 (right) on 4–13 May 2016.

which is not captured by MERRA-2. One reason is that the data assimilation in MERRA-2 is limited at these altitudes, what
5 results in less captured tidal features in MERRA-2 compared to the lidar data. Another reason is the sponge layer as upper boundary in MERRA-2, which also damp wave amplitudes near the model top. But nevertheless, the total tidal variation from the lidar temperatures shown in the figure is in phase with the tidal variation obtained by MERRA-2 almost over the whole altitude range below 60 km. For a closer look, the tidal fluctuations of both data sets are shown together for one particular altitude of 50 km in Fig. 3. The data sets agree very well in phase and even the magnitude of the tidal temperature fluctuations
10 is comparable. Aside of the differences mentioned above, the results demonstrate that MERRA-2 provides suitable temperature data for a comparison to lidar data regarding tidal phase studies. From this we assume that the MERRA-2 wind is also related to the realistic winds, although we have not tested this.

The wind data from MERRA-2 are treated in the same way as the temperature data and tidal wind fluctuations are separately calculated for the zonal and the meridional winds. The tidal wind fluctuations are shown in Fig. 4 for both wind components





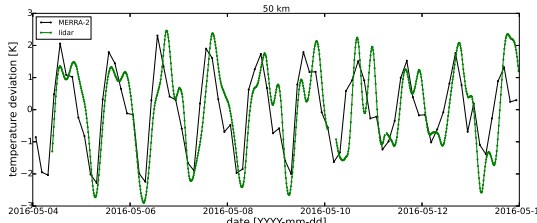

**Figure 3.** Comparison of tidal temperature fluctuations from the lidar (green) and MERRA-2 data (black).

over the same sounding period and the same altitude range as the lidar data. Clear large scale structures are visible in the

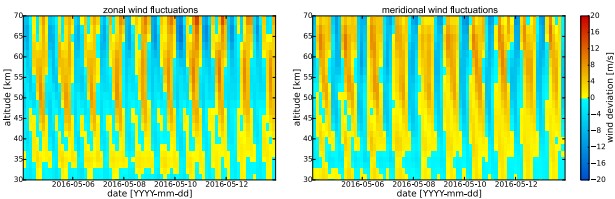

**Figure 4.** Tidal fluctuations derived from the zonal (left) and meridional winds (right) given by MERRA-2 for the same period in May 2016.

data from both wind components. The total tidal wind fluctuations are mostly dominated by a diurnal signature, especially this is the case for the meridional tidal wind fluctuations. The same behavior was already seen in the temperature fluctuations induced by tides. Below 45 km as well as above 65 km also a semidiurnal component occurs in both wind components. While the temperature fluctuations induced by tides are in the order of a few Kelvin (relative temperature change of ∼1%), the wind fluctuations are in the range of 10 m/s (relative wind change of ∼30%), so the response of the wind on tides is larger compared to the temperature response. The results for the concrete tidal amplitudes and phases will be presented in the next subsection.

### 3.2 Amplitudes and Phases

For the determination of the phase relation between the tidal temperature and the tidal wind fields, we first compare the amplitudes and phases as a mean over the 10 days in May 2016 in an altitude range between 30 and 70 km. The amplitudes for the diurnal, semidiurnal and terdiurnal tide are shown in Fig. 5 respectively. The amplitudes from the temperature field are derived using the lidar data as well as the data from MERRA-2. In principle, both data sets reveal similar amplitudes for all three tidal temperature components. Larger differences only occur for the terdiurnal component at altitudes above 55 km, where the amplitudes from the lidar temperature show a strong increase in contrast to the amplitudes from the MERRA-2 temperatures.

As already seen from the fluctuations themselves, the amplitudes derived from the wind fields are much larger compared to those from the temperature field. A direct comparison of the amplitudes is made from the relative change according to the mean background temperature and the mean background wind. Such comparison reveals a tidal temperature response of





only ∼1% and a tidal wind response of ∼30%. This strong wind response is in particular the case for the diurnal component above 45 km altitude, while e.g., the terdiurnal component is comparably weak in the winds over the whole altitude range. For this component there is even a stronger increase of the amplitude from the temperature visible for altitudes above 60 km compared to those from the winds, although the values are underestimated in this altitude range. But however, the amplitudes are much smaller than from the diurnal and semidiurnal component. The diurnal wind tide is the dominating tidal component

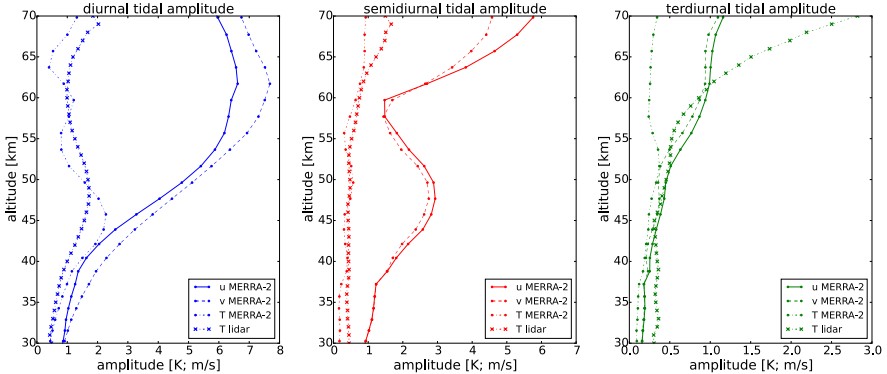

**Figure 5.** The diurnal (blue), the semidiurnal (red) and the terdiurnal (green) tidal amplitudes derived from the lidar temperature field (dotted line with cross marks), from the MERRA-2 zonal (solid line) and from the meridional wind field (dashed line) as mean over the entire days of the sounding period in May 2016. For comparison also the tidal amplitudes from the temperature field of MERRA-2 is shown (dotted line with point marks).

in an altitude range between 45 and 65 km, while the semidiurnal wind tide shows a strong decrease around 60 km, this is completely different for temperature tides. Here, the amplitudes of the diurnal temperature tide show an increase up to almost 2 K at 50 km altitude, while the amplitudes fall off again to values around 1 K between 60 and 65 km. This behavior is also evident for the amplitudes derived from the MERRA-2 temperatures. As the results of the temperature tides are comparable from the lidar and MERRA-2, further descriptions of the temperature tide always refer to the lidar data. The semidiurnal temperature tide instead is almost constant with altitude and shows only a slow increase up to 2 K above 55 km. Furthermore, the strength of the diurnal wind tide is getting weaker above 60 km, this is the altitude range, where the semidiurnal wind tide shows an increase. For the MLT region it is already known that the semidiurnal tide is the dominating tide, what is also indicated from the study here.

The mean tidal phases are shown in Fig. 6 in the same way as the amplitudes for the diurnal, the semidiurnal and the terdiurnal component derived from the temperature field and the wind fields, respectively. The mean phase of the diurnal tidal signature is nearly constant with altitude from the lidar temperatures as well as from the winds. Between the zonal and meridional tidal wind component, a constant phase shift of around -6 h is visible for the whole altitude range above 45 km. This corresponds to a phase shift of -90° for the diurnal tide. The phase shift between the diurnal temperature tide and the diurnal zonal wind tide is almost zero up to an altitude range of 55 km. For larger altitudes as well as for altitudes lower than 35 km the phase of the temperature tide is decreasing because of the small amplitude of the diurnal signature in the temperature.





These are regions where a harmonic analysis is not well determined. The results for the semidiurnal component look different

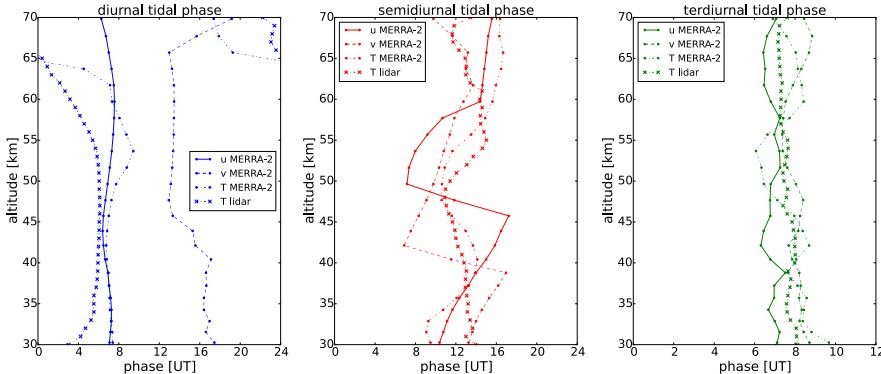

**Figure 6.** The diurnal (blue), the semidiurnal (red) and the terdiurnal (green) tidal phases derived from the lidar temperature field (dotted line with cross marks), from the MERRA-2 zonal (solid line) and from the meridional wind field (dashed line) as mean over the entire days of the sounding period in May 2016. Please note the different x-scale for the terdiurnal tidal phase. For comparison also the tidal phases from the temperature field of MERRA-2 are shown (dotted line with point marks).

compared to the diurnal tidal phases. The semidiurnal tidal phases of both wind components increase with larger altitudes, but both wind components show a phase jump around 40 and 47 km for v and u, respectively. These jumps are the explanation for the visible semidiurnal tidal component in the total tidal wind fluctuations at lower altitudes (see Fig. 4). Due to the changing

phase, the semidiurnal variation is shifted and less visible. The meridional semidiurnal wind tide leads the zonal one with 3 h, what corresponds to a phase shift of again 90° in an altitude range between 30 and 40 km and between 50 and 57 km. The information from larger altitudes are not taken into account due to their reduced reliability. Between 40 and 50 km the phase shift of zonal and meridional semidiurnal tidal winds is almost 12 h, what means both components are in phase. The phase of the semidiurnal temperature tide shows a slow decrease from 14 to 12 h up to 50 km altitude. Beyond that, the phase increases

again and stays constant at 15 h above 55 km. A fixed phase relation to the wind components can only be given between 50 and 57 km altitude with 6 h (or 180°) between the temperature tide and the meridional wind tide.

The mean phases of the terdiurnal tidal component are almost constant with altitude from the temperature as well as from the wind field. The phase shift between the zonal and meridional wind tide slightly varies around -2 h (-90°). Thereby, the meridional wind leads the zonal wind tide in the same way as seen for the other tidal components. At 39 and 55 km the phases

of all components become equal. Especially for lower altitudes, the amplitude of the terdiurnal wave component is very small, which causes larger difficulties and uncertainties in the fitting algorithms yielding in a zero phase shift between the temperature and the wind tide. In general, the mean phases reveal a fixed phase relation between both wind components and a semi-fixed phase relation between the temperature and the zonal wind tide. For the sake of completeness, a comparison between the temperature tide derived from the lidar and from MERRA-2 was also done based on mean amplitudes and mean phases. In

principle, both temperature tides show the same behavior with an almost zero phase shift, what makes a comparison of these data sets possible.

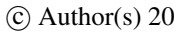



As these results are only based on mean values, the next step now is to show the time-resolved amplitudes and phases derived with the adaptive spectral filtering. Each tidal component is separately shown in the following figures, but the amplitudes and phases from the temperature tide, from the zonal and the meridional wind tide are shown together for a better direct comparison. First of all, we focus on the diurnal tidal components, which is shown in Fig. 7. The amplitudes from both wind components

5 grow with larger altitudes during the whole sounding period (Fig. 7 e, g), what is excepted due to the decreasing air density and the conservation of energy. Contrary to this, the tidal amplitude from the temperature data shows a saturation and even a

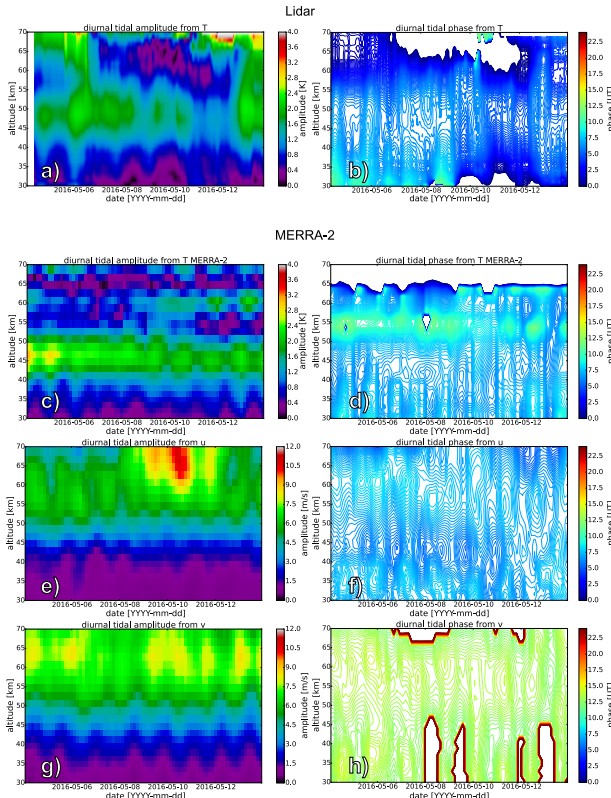

**Figure 7.** Comparison of the time-resolved diurnal tidal amplitudes (a, c, e, g) and phases (b, d, f, h) derived from the lidar/MERRA-2 temperature field and the MERRA-2 zonal and meridional wind field in May 2016.

decrease above 55 km. This is evident in the lidar data as well as in the MERRA-2 data (Fig. 7 a, c). While the amplitude also decreases with time from the lidar temperatures and even for the MERRA-2 temperature data, the wind tides stay constant with time except for a small periodic oscillation. The amplitude of the meridional wind tide is larger than of the zonal wind tide by a factor of around 1.3 for altitudes below 60 km. From the time-resolved phases (Fig. 7 f, h) it can be clearly seen, that the

10 phase shift between the zonal and meridional wind is around -6 h (the meridional wind tide leads the zonal wind tide) during the whole sounding. The diurnal temperature tide shows a similar phase progression like the diurnal zonal wind tide (Fig. 7



b, d, f). Differences from this phase relation mainly occur for the lidar data for altitudes larger than 60 km as well as at low altitudes around 35 km, while the MERRA-2 temperature tide matches the phase of the zonal wind tide. These differences can be explained by a different temporal and vertical resolution of the data sets and limitations in data assimilation.

The time-resolved amplitudes and phases of the semidiurnal tidal signature are shown in Fig. 8 in the same way as for the diurnal tidal component. As already seen from the diurnal amplitudes, also the semidiurnal tides are stronger visible in the

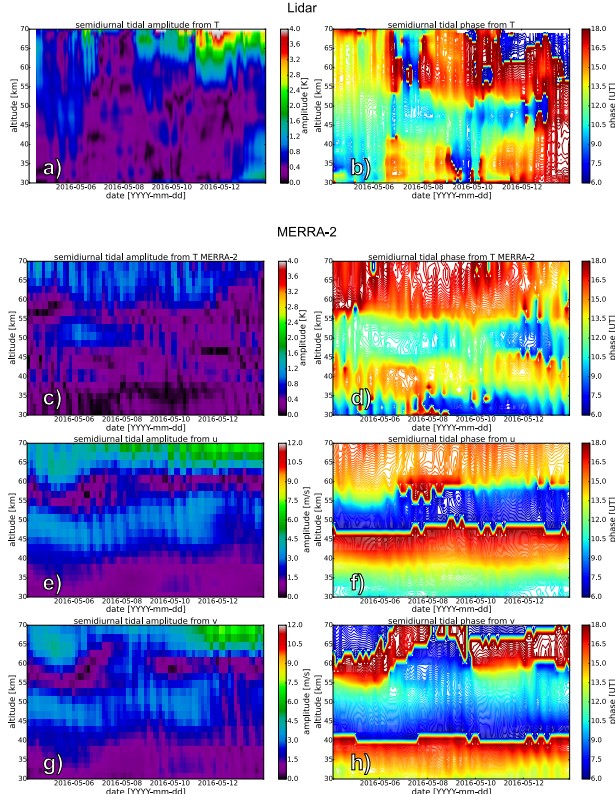

**Figure 8.** Same as Fig. 7 for the semidiurnal tidal signature.

wind data than in the temperature data. But now for the semidiurnal tide, the amplitude of the zonal wind tide is slightly larger than from the meridional wind tide. The ratio of the meridional and the zonal wind amplitude is around 0.8. Further differences to the diurnal tidal component are seen from the altitude growing and the temporal variation. The semidiurnal tidal amplitude from the temperature data is increasing with altitude (Fig. 8 a), while this is not the case for the wind components. Here, the

10  amplitudes drop down between 55 and 60 km (Fig. 8 e, g). In this altitude range also the phases are not stable due to too small amplitudes of the semidiurnal component. But nevertheless, in between where no phase jumps occur, the phases are slowly varying with time. The phase difference between the zonal and the meridional wind tide is changing over the altitude. There




are ranges, where both components are almost in phase due to a phase jump or where both components have a phase shift of
-3 h. This is visible e.g., at 45 km (phase shift of 10 h, which means almost in phase) or at 35 km (phase shift of -90°).

The results for the terdiurnal tidal component (Fig. 9) look even more complicated as the overall variation is smaller than for
the diurnal and semidiurnal component and therefore, a robust phase determination is more difficult. Due to this circumstance

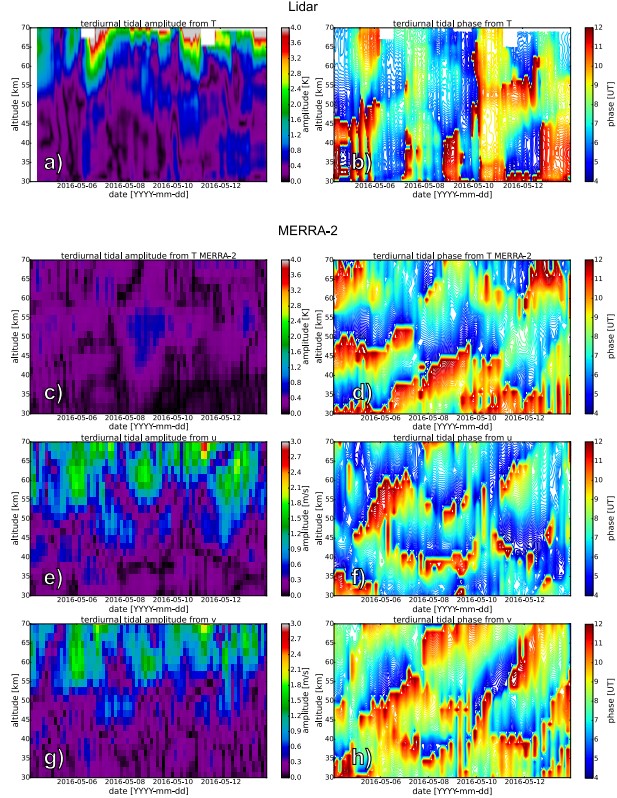

**Figure 9.** Same as Fig. 7 for the terdiurnal tidal signature. Note the different color scale for the phases compared to Fig. 7 and 8.

5   a phase determination has a larger error. The amplitude of the zonal wind tide is a little bit larger than from the meridional wind
tide (Fig. 9 e, g). The ratio of the meridional to the zonal amplitude is about 0.7. But in general, the same temporal variation is
visible for both wind tides. The overall phase variation is around ±4 h, what causes the impression of more phase jumps over
the time than for the other tidal components as the overall variation was smaller for the other components. The phase difference
between the zonal and meridional terdiurnal tide is constant at about -2 h during the whole sounding. So a fixed phase shift of
10   -90° between the wind components is still valid. The terdiurnal temperature tide is in phase with the zonal wind tide for certain
times and altitudes, especially for the first days of the sounding. During the last days the phase of the temperature tide is shifted
to the meridional wind tide.



A comparison between the tidal results derived from the lidar temperatures and the MERRA-2 temperatures shows in general a good agreement. The decrease of the diurnal tidal amplitudes with time is also in MERRA-2 visible, but not as strong as in the lidar data (Fig. 7 a, c). However, the amplitudes are in the same range for both data sets. The temporal variation of the amplitudes seen in the lidar data is not fully captured by MERRA-2, but the basic features are comparable, e.g., amplitude

saturation of the diurnal tide above 55 km. The time-resolved phases of the lidar temperature tides and the MERRA-2 temperature tides roughly agree over the 10 days in May 2016. The phase progression in general indicates that the tidal phase is not constant over time as even in MERRA-2 data the phases show a small variation with time. This needs to be further investigated during the whole year or at least for other seasons and it will be done in another study as this is out of the scope of this paper.

### 3.3   Tidal modes from MERRA-2 global fields

MERRA-2 also provides global fields of temperature and winds in the atmosphere. Those fields are used to determine which tidal modes are observed at the ground with local lidar measurements. Therefore, the global fields are filtered by applying the adaptive spectral filter including the zonal wave number and longitude (eq. 3) to the data. The resulting amplitudes for the migrating diurnal and semidiurnal tide (DW1 and SW2) are shown in Fig. 10 and 11 respectively. The nomenclature follows Oberheide et al. (2015). The diurnal migrating tide (DW1) shows in principle the same strength and the same altitudinal

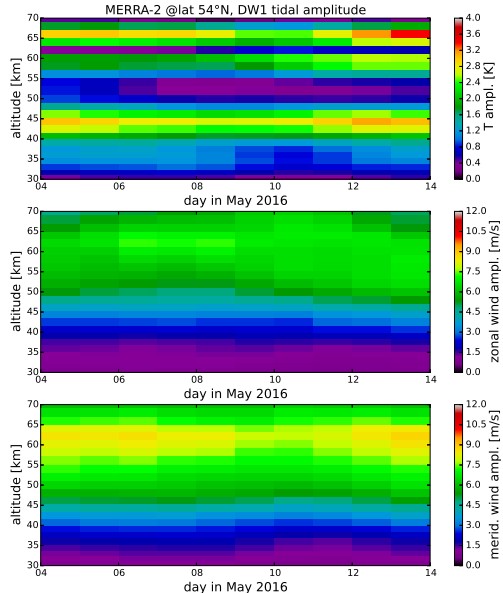

**Figure 10.** DW1 amplitude in temperature, zonal and meridional wind from global MERRA-2 fields at the latitude of 54° N.

structure from the global fields as from the local perspective (Fig. 7). There is a localized maximum of amplitudes in the




temperature field at altitudes around 45 km, while the tidal wind amplitudes show an increase with altitude instead. A similar agreement to the local data is found for the semidiurnal migrating tide (SW2). Due to the coarser temporal resolution of the global fields, the short-term variation of both tidal components is not captured. However, the good agreement of these two single tidal components to the local data indicates that the observations are dominated by single tidal waves.

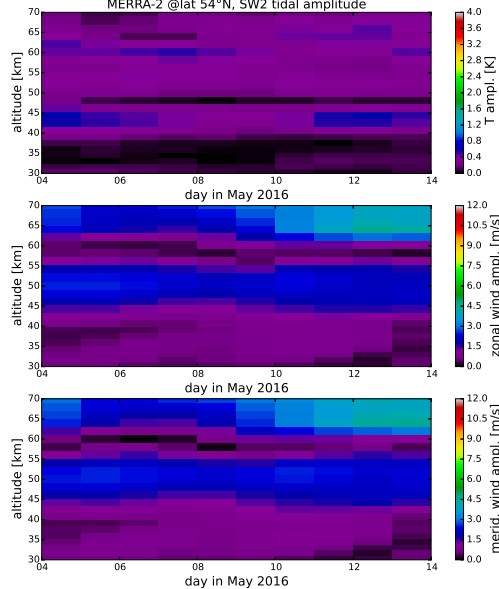

**Figure 11.** SW2 amplitude in temperature, zonal and meridional wind from global MERRA-2 fields at the latitude of 54° N.

5     A comparison of these two components to other potential tidal components derived for the location of the lidar station at Kühlungsborn during May 2016 is shown in Fig. 12 at an altitude of 45 km. It is clearly visible that the DW1 is the most

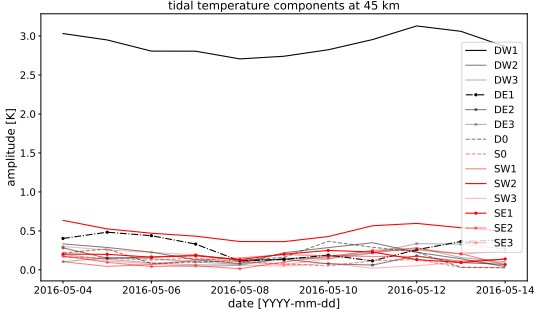

**Figure 12.** Tidal temperature modes at 45 km altitude from global MERRA-2 fields at the latitude of 54° N.





dominating tidal component over the whole time period. The amplitudes of the non-migrating tides as well as of the zonally symmetric tide are smaller than 1 K and therefore insignificant compared to the DW1 tide. The same appears for the semidiurnal tide, although this component is generally weaker than the diurnal component. The amplitudes of potential other diurnal and semidiurnal tidal components including the zonally symmetric and non-migrating tides separated for each wave number are

shown in the Appendix.

## 4   Discussion

This paper addresses the relationship between tidal fluctuations observed in temperatures and winds in the stratosphere and mesosphere during 10 days in May 2016 using a ground-based, daylight-capable RMR lidar located at Kühlungsborn (54° N, 12° E) together with reanalysis data provided by MERRA-2. A similar study has been done by She et al. (2016) focusing on

the MLT region at Ft. Collins (41° N, 105° W). They used a Na lidar combined with model predictions from the Climatological Tidal Model of the Thermosphere (CTMT) based on the Hough Mode Extension (HME) technique to investigate the polarization relation of the diurnal and semidiurnal tide. They have derived the polarization relations for tides from the primitive equations for large scale waves (Holton, 2004). Following them, the derived polarization relation between the temperature and the zonal wind tide is equal to that for the low-frequency gravity waves (Hu et al., 2002). The polarization relation between the

meridional and the zonal wind tide becomes even simpler by assuming just a single migrating wave (She et al., 2016):

$$\widetilde{v} = -\mathrm{i}\frac{f}{\widehat{\omega}}\widetilde{u} \qquad (4)$$

with $f$ as Coriolis frequency, and $\widehat{\omega}$ as intrinsic frequency of the tidal component. With this equation a ratio of the tidal meridional and zonal wind amplitudes is theoretically known. For a latitude of 54° N this ratio is 1.6 and 0.8 for the diurnal and semidiurnal tidal component, respectively. The results of our analysis are very good in agreement to these values. This

suggests that the tidal waves from MERRA-2 are dominated by the migrating tide and non-migrating tidal modes only show a negligible contribution during the evaluated period. Additionally, the global analysis of MERRA-2 temperature and wind fields also showed a dominance of the diurnal and semidiurnal migrating tide (see Fig. 10, 11 and 12), which confirms the result derived using the ratio of zonal and meridional tidal wind amplitudes. Smaller differences to the values expected are basically due to the simplifications in deriving the relation between the meridional and zonal tide as there is neither a latitudinal

background temperature gradient nor a dissipation included. Additional to this, the altitude range covered with the lidar and MERRA-2 includes source regions of tidal excitation as well as regions where trapped tidal modes can exist (Forbes and Garrett, 1979; Forbes and Hagan, 1988; Yuan et al., 2010; Kopp et al., 2015). But however, the general behavior of the tidal signatures is in agreement to the theoretical expectations.

The wind tides from MERRA-2 and the temperature tide from the lidar showed that there is a fixed phase relation between

both wind components and a semi-fixed phase relation between the temperature tide and the winds. Typically, observations are limited either to a detection of temperatures or winds. Tidal information based on both quantities have been simultaneously derived in the past from model outputs, e.g., from the Kühlungsborn Mechanistic General Circulation Model (KMCM) by



Becker (2017). This study addresses the heat budget of the mesosphere and lower thermosphere regarding upward propagating thermal tides. Therefore, also the mean wind tides were shown together with the mean temperature tide for January and July at 52° N. From the July data, it is clearly visible that the mean zonal wind tide is phase shifted with -90° to the mean meridional wind tide. This is exactly what we have observed from the tidal analysis of the winds provided by MERRA-2. In Becker (2017)

the mean temperature tide from KMCM is shown as total tidal variation. Hence, a direct comparison of the amplitudes for each tidal component is not suitable. But from the KMCM temperature tide itself, an altitude dependence of the dominating tidal component is observable. Above 70 km Becker (2017) found a stronger semidiurnal component compared to the lower altitudes, where a diurnal component is dominating. This transition from a diurnal to a semidiurnal tidal component is also indicated in the lidar data above an altitude of 60 km above Kühlungsborn from this study (see Fig. 2). Such a development

is also known from other studies (e.g., Smith, 2012; Pokhotelov et al., 2018). Aside from the fact, that the amplitudes from KMCM cannot be compared directly to our data, a general phase relation between the temperature tide and the wind tide from KMCM can still be derived for comparison. The temperature tide follows mainly the mean zonal wind tide in KMCM for altitudes below 50 km, which is confirmed by our data. Above 50 km altitude, a phase shift of 180° occurs in the KMCM data. This agrees to the semidiurnal tidal component from our study and also the diurnal tidal component shows a larger phase shift

to the zonal wind tide from MERRA-2 above 50 km.

The explanation for the changing phase with growing altitude is directly written in the derived phase relation between the temperature tide and the zonal wind tide provided by She et al. (2016). The relation of both quantities is highly depending on the vertical wavelength and the period. The latter is fixed for each tidal component, but the vertical wavelengths are not constant with altitude. The main reason for this is wind filtering and dissipation. The altitude dependence of the vertical wavelength or

the vertical wave number can be in principle determined from the phase profiles. But the time-resolved phases showed a large temporal variability. This temporal tidal phase variation due to an intermittency in the tidal excitation is taken into account with the ASF, while other analysis methods assume a constant phase at least for the migrating tide. This has an impact on the interpretation of the significance of migrating and non-migrating tides. But solely from the observed temporal variability, we claim that this behavior has to be taken into account when tidal information is calculated from e.g., satellite data. As those

instruments need large time intervals in covering one day at one location, they typically assume a constant phase of the tides. Our investigations address this issue and demonstrate even for the diurnal tidal component, that the phase is not stable over 10 days of observation. Therefore, we suggest that the phase shows a drift of several hours over the whole year and further phase jumps occur during special occasion. Using MERRA-2 wind data next to the available lidar data, we will be able to provide further information about tides and the phase relations between the temperature and the wind tide during different seasons,

which will enhance our knowledge about the dynamics in the atmosphere. This will be addressed in a future paper.

## 5    Conclusions

The phase relation from tidal signatures in temperatures and winds using lidar and reanalysis data was studied to reveal if both tidal fields have a fixed phase relation. Therefore, an outstanding temperature data set obtained by the RMR lidar located at




midlatitudes in Kühlungsborn was used in combination with reanalysis data of MERRA-2. This provides the necessary wind data in the same altitude range as the lidar data. The main results are the following:

1) MERRA-2 temperatures show in principle good agreement to the lidar temperatures. Tidal amplitudes and phases are comparable. Main differences occur for the terdiurnal tidal component. Due to the higher temporal and vertical resolution of the lidar, this instrument captures the terdiurnal component much better than MERRA-2. However, additional wind information provided by MERRA-2 are suitable to be used for a comparison of temperature and wind tides.

2) There is a fixed phase relation between the zonal and meridional tidal wind fluctuations. The meridional wind fluctuations lead the zonal wind fluctuations with a phase shift of 90°. This appears to be constant for all altitudes.

3) There is no fixed phase relation between the temperature and wind fluctuations over the whole altitude range. But nevertheless, a phase relation is existing within certain altitude levels. The phase shift between tidal temperature fluctuations and wind fluctuations is strongly dependent on altitude and period. The relation follows the polarization relation of low-frequency gravity waves. If the vertical wavelength and the period is extracted from the data, a known tidal field can be transferred into another tidal field.

4) The global analysis of MERRA-2 confirms that the tidal temperature and wind field locally seen is dominated by the migrating diurnal and migrating semidiurnal tide, if an intermittency of the wave field is taken into account, while other tidal components are negligible.

This analysis complements studies using resonance lidar measurements combined with modeling results in the MLT region. This knowledge will be used in future to deduce an unknown tidal field from another observed tidal field and will enhance the data coverage from single observations.



## Appendix A: Tidal components from the global MERRA-2 fields

In addition to Fig. 10 and Fig. 11 we provide here the results for the stationary diurnal and semidiurnal tide (D0, S0) as well as for the westward and eastward propagating non-migrating tidal components (DW2, DW3, SW1, SW3, DE1, DE2, DE3, SE1, SE2, SE2).

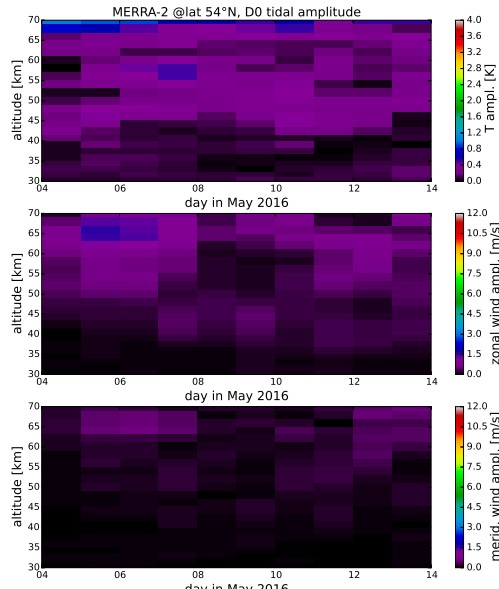

**Figure A1.** D0 amplitude in temperature, zonal and meridional wind from global MERRA-2 fields at the latitude of 54° N.



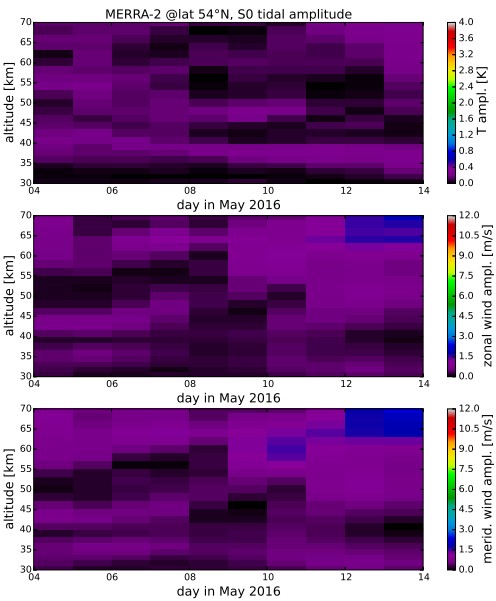

**Figure A2.** S0 amplitude in temperature, zonal and meridional wind from global MERRA-2 fields at the latitude of 54° N.

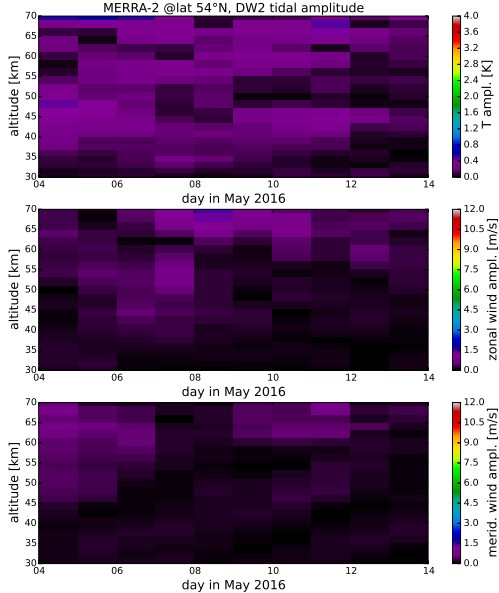

**Figure A3.** DW2 amplitude in temperature, zonal and meridional wind from global MERRA-2 fields at the latitude of 54° N.



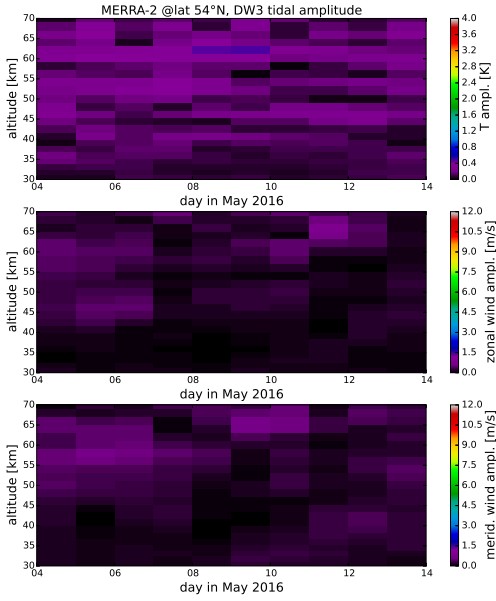

**Figure A4.** DW3 amplitude in temperature, zonal and meridional wind from global MERRA-2 fields at the latitude of 54° N.

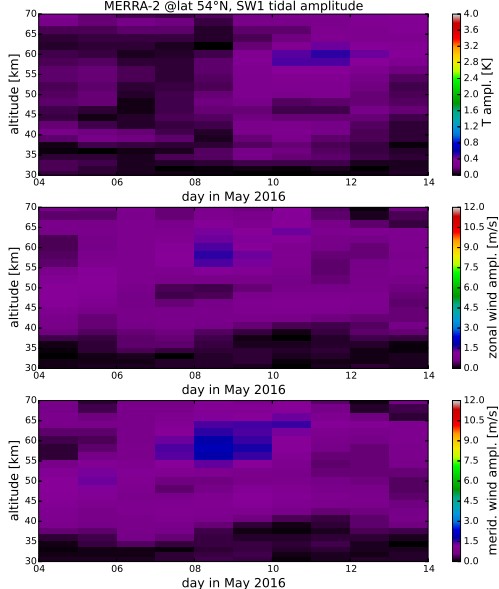

**Figure A5.** SW1 amplitude in temperature, zonal and meridional wind from global MERRA-2 fields at the latitude of 54° N.





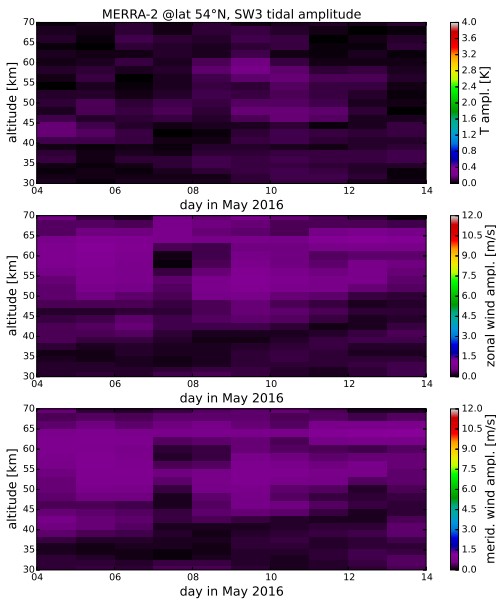

**Figure A6.** SW3 amplitude in temperature, zonal and meridional wind from global MERRA-2 fields at the latitude of 54° N.

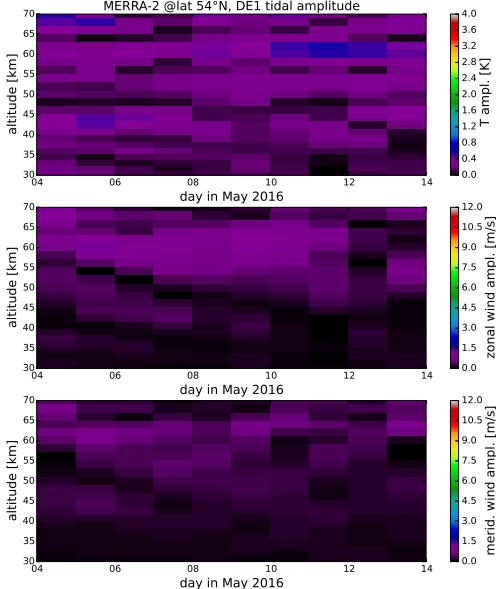

**Figure A7.** DE1 amplitude in temperature, zonal and meridional wind from global MERRA-2 fields at the latitude of 54° N.





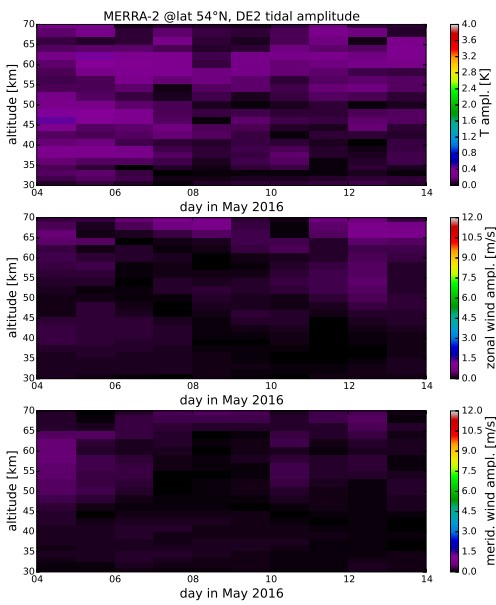

**Figure A8.** DE2 amplitude in temperature, zonal and meridional wind from global MERRA-2 fields at the latitude of 54° N.

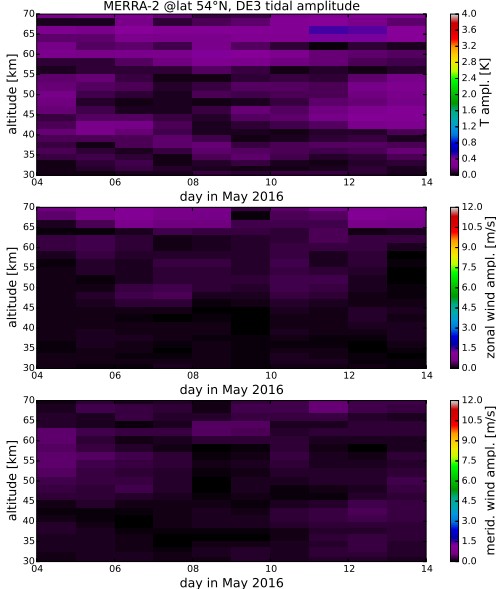

**Figure A9.** DE3 amplitude in temperature, zonal and meridional wind from global MERRA-2 fields at the latitude of 54° N.





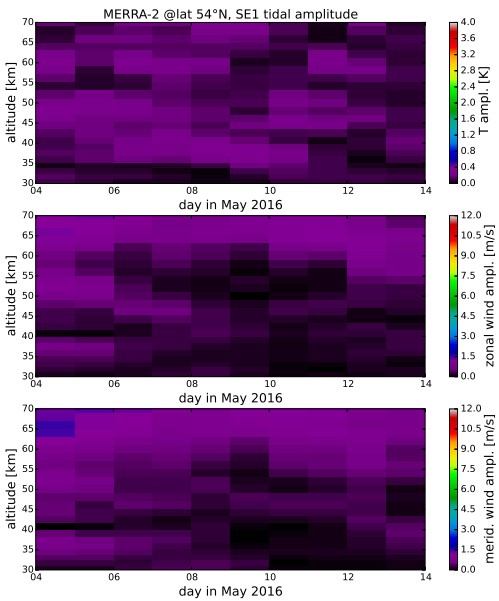

**Figure A10.** SE1 amplitude in temperature, zonal and meridional wind from global MERRA-2 fields at the latitude of 54° N.

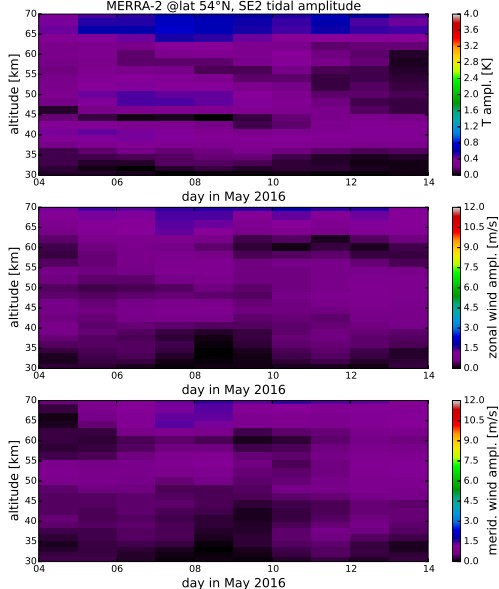

**Figure A11.** SE2 amplitude in temperature, zonal and meridional wind from global MERRA-2 fields at the latitude of 54° N.





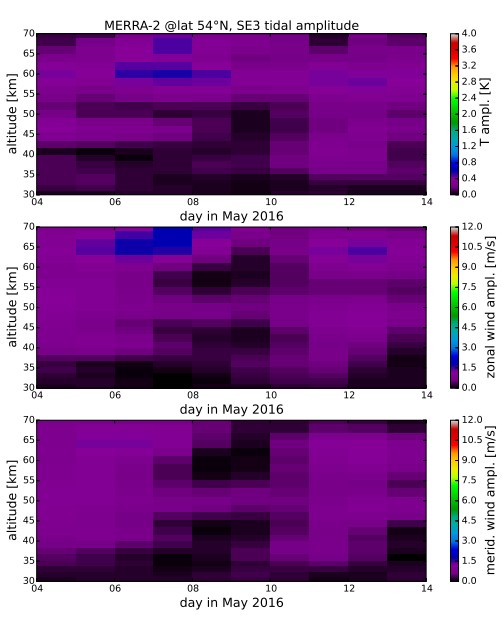

**Figure A12.** SE3 amplitude in temperature, zonal and meridional wind from global MERRA-2 fields at the latitude of 54° N.



*Author contributions.* KB calculated the lidar temperature data, visualized and interpreted the temperature and wind data from the lidar and MERRA-2 and drafted the first version of the manuscript. GS wrote the data analysis code, provided the MERRA-2 data, performed the analysis and discussed the findings.

*Competing interests.* The authors declare that they have no conflict of interest.

5  *Acknowledgements.* We gratefully acknowledge Michael Gerding for his helpful comments and for being the responsible scientist of the RMR lidar system at IAP. We also thank Maren Kopp and Josef Höffner for their help in the installation of the daylight capable RMR lidar as well as Michael Priester and Torsten Köpnick for the maintenance and operation of the RMR lidar system. We also acknowledge all students for helping in lidar operation. We thank the tidal matrix group at IAP for fruitful discussions. The MERRA-2 data provided by the Global Modeling and Assimilation Office (GMAO) at NASA Goddard Space Flight Center were acquired through the NASA GES DISC
10  online archive. The lidar data is locally archived at the IAP and available upon request from the author. This work was partially supported by the Deutsche Forschungsgemeinschaft (DFG, German Research Foundation) under the SPP1788 (DynamicEarth) project DYNAMITE (CH1482/1-1) as well as under project LU1174/8-1 (PACOG) of the research unit FOR1898.





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
