# Peer review of "On the evaluation of the phase relation between temperature and wind tides based on ground-based measurements and reanalysis data in the middle atmosphere"

_Annales Geophysicae, 2019_

## Referee Comment (RC1) · Anonymous Referee #1 · 10 Apr 2019

This manuscript describes temperature tides and their connection to the wind behaviour in the middle atmosphere using ground-based observation and reanalysis MERRA 2. Authors give very clear and useful results. I have only two minor question which should be implemented into discussion part and I can reccommend this paper for publication. 1) MERRA 2 provide 3 hourly data up to 0.1 hPa. For diurnal and semidurnal tides is should be ok. But I am not sure if for the terdiurnal tide especially at higher altitudes MERRA 2 data are reliable for this kind of studies. It should be at least mentioned in the paper. 2) I suppose that data from lidar observations is

included into MERAA 2 reanalysis assimilation proccess. Could it affect the results or not? Especially if you show the comparison between MERRA 2 and lidar observation.

---

## Referee Comment (RC2) · Anonymous Referee #2 · 2 May 2019

The paper by Baumgarten and Stober investigates the temporal variation of tides and their phases by applying constrained fits to a unique data set of continuous lidar data. With these they validate tidal signatures in MERRA-2 up to an altitude of somewhat below 60km altitude. MERRA data on the other hand are used to investigate the phase relations between the different variables. These findings are new and interesting, the evidence presented supports the conclusions and the paper is well to read. The paper is therefore recommended for publication in Annales Geophysicae. One result is surprising (at least to me): the phase of the diurnal tide is constant over an altitude of

30km. I would like ask the authors at least to comment on this in the paper (see my minor comment below).

Minor comment:

The data analysis has been thoroughly performed, so we have to believe the phases as given in Figure 6 and Figure 7. The phase does not alter with altitude and for this there is also very little change in time. On the other hand you would expect a vertical wavelength of roughly 30km (cf. quotations in your text). That would mean that the phase runs one time over $360°=24$ hours in that altitude range - similar as the semi-diurnal tide does. This is an unexpected behavior and should be commented on as such. And also the terdiurnal tide: There is phase variation with altitude, if you look into the time resolved data of Figure 9; this is also an interesting result that the expected phase variation with altitude from the temporally resolved data disappears in the mean profiles.

It would be nice, but may be exceeding the scope of this paper, if you could use MERRA data to reveal the global structure of the migrating tide. Where in the Hough mode structure is Kühlungsborn located? Probably close to a node of the Hough mode for temperature? This would explain the relatively small amplitudes in T compared to winds? How does vertical phase structure (in a zonal mean sense) at Kuehlungsborn-latitude compare to that at the equator? As said, maybe too much for the paper but it could help to understand things.

Specific comments and technical corrections:

P1L24 ... for semidiurnal component and so on are ...

P3L2 trapped tidal mode in SABER? in MERRA?

P3L3 please reformulate follow similiar as GWs poloarization relations?

P3L13 define s in k and motivate why is there a suffix s at the scale height

P3L19 Probably. The only one which I could think of is CORAL at Rio Grande. (just check, no action required, if still holds)

P3L28 as well as fluctuations/variations due to GWs

P5L5 ... but ?here?in that work? without ...

P5L14 variety -> variance

P5L19 somehow the assumption and the regularization seem to be described twice - choose the more precise formulation

P5L23 Studies such as ...

P5L27 if you filter them out that could be a self-fulfilling - but I think you are right

P5L29 It appears -> It turns out ??

P6L7 result section

P6L10 time resolved *what*

P6L29 For the left column this could be either coarser vertical resolution or MERRA's horizontal resolution which is filtering in a different dimension than the ones shown/discussed here.

P7L1 MERRA data are snap-shot data? Then only the temporal sampling is coarser, but the temporal resolution should be sufficient.

P7l6 damps

P8l6 as far as it makes sense to compare the values of different units - this becomes a meaningful result, if you set it in relation to the expected relations for the maxima of the Hough modes

P9L21 Why should the amplitude influence the phase? Is there a second mode competing? Are you just speculating about a higher uncertainty? -> P10L1 o.k., but you

have never discussed limits so far.

P10L3 My interpretation of the phase jump would have been that the phase cyclically continues in your fitting interval (the phase jumps are 12h = 360°). Whether the point around 12 UT is a real independent point may depend on your fitting technique. Would your technique allow the phase to continuously incerase even beyond 360°? O.k. the fact that the phase shifts with respect to the diurnal tide makes the semidiurnal tide visible in F4. The main point should be still the relative amplitude ratio.

F7-F9: Why are you using contour lines for the phases? That makes the plots very hard to read.

P16L18 in very good agreement

---

## Author Comment (AC1) · 22 May 2019

**Author's response on "On the evaluation of the phase relation between temperature and wind tides based on ground-based measurements and reanalysis data in the middle atmosphere" by Kathrin Baumgarten and Gunter Stober.**

**Anonymous Referee #1**

This manuscript describes temperature tides and their connection to the wind behaviour in the middle atmosphere using ground-based observation and reanalysis MERRA 2. Authors give very clear and useful results. I have only two minor question which should be implemented into discussion part and I can reccommend this paper for publication.

*We thank the reviewer for the comments. Answers are given below in italics.*

1) MERRA 2 provide 3 hourly data up to 0.1 hPa. For diurnal and semidurnal tides is should be ok. But I am not sure if for the terdiurnal tide especially at higher altitudes MERRA 2 data are reliable for this kind of studies. It should be at least mentioned in the paper.

*We agree to the raised comment. MERRA-2 data resolution limites the representation of the terdiurnal tidal component in the results, while the lidar data resolution is better. We have mentioned this in the paper in conclusions (P18L15-16).*

2) I suppose that data from lidar observations is included into MERRA 2 reanalysis assimilation proccess. Could it affect the results or not? Especially if you show the comparison between MERRA 2 and lidar observation

*MERRA-2 mainly assimilates data from satellites. A list of instruments that enter the data assimilation can be found in Gelaro et al., 2017, JCli. There is no data assimilated from the Kborn lidar system used in this study. Definitely assimilated measurements are not suitable to provide an independent cross comparison to the model state vectors (temperature, winds).*

---

## Author Comment (AC2) · 22 May 2019

**Author's response on "On the evaluation of the phase relation between temperature and wind tides based on ground-based measurements and reanalysis data in the middle atmosphere" by Kathrin Baumgarten and Gunter Stober.**

**Anonymous Referee #2**

The paper by Baumgarten and Stober investigates the temporal variation of tides and their phases by applying constrained fits to a unique data set of continuous lidar data.With these they validate tidal signatures in MERRA-2 up to an altitude of somewhat below 60km altitude. MERRA data on the other hand are used to investigate the phase relations between the different variables. These findings are new and interesting, the evidence presented supports the conclusions and the paper is well to read. The paper is therefore recommended for publication in Annales Geophysicae. One result is surprising (at least to me): the phase of the diurnal tide is constant over an altitude of 30km. I would like ask the authors at least to comment on this in the paper (see my minor comment below).

*We thank the reviewer for the mindful and constructive comments. We revised the manuscript with regard to your comments. Detailed answers are given below. The line numbers for the changes refer to the manuscript with marked changes.*

Minor comment:

The data analysis has been thoroughly performed, so we have to believe the phases as given in Figure 6 and Figure 7. The phase does not alter with altitude and for this there is also very little change in time. On the other hand you would expect a vertical wavelength of roughly 30km (cf. quotations in your text). That would mean that the phase runs one time over $360°=24$ hours in that altitude range - similar as the semi-diurnal tide does. This is an unexpected behavior and should be commented on as such. And also the terdiurnal tide: There is phase variation with altitude, if you look into the time resolved data of Figure 9; this is also an interesting result that the expected phase variation with altitude from the temporally resolved data disappears in the mean profiles. It would be nice, but may be exceeding the scope of this paper, if you could use MERRA data to reveal the global structure of the migrating tide. Where in the Hough modestructure is Kühlungsborn located? Probably close to a node of the Hough mode for temperature? This would explain the relatively small amplitudes in T compared to winds? How does vertical phase structure (in a zonal mean sense) at Kuehlungsborn-latitude compare to that at the equator? As said, maybe too much for the paper but it could help to understand things.

*From a mean perspective it is known that a propagating wave would lead to a tilted phase line over the time, from which the phase velocity and the vertical wavelength could be extracted. There are some regions where this is indeed the case. We know from simulations such as by Lindzen, 1967, Quart. J. Roy. Meteorol. Soc., that at low latitudes there is a downward progression of the phase, while at higher latitudes there is a relative constancy of the phase with altitude. Therefore, our results of the phase with altitude are not unlikely. In general it is possible to investigate the global tidal structure from MERRA-2 data. We used these data to identify the dominating tidal component, but just for the location of Kuehlungsborn. However, looking at the global structure in detail is out of the scope of this paper.*

Specific comments and technical corrections:

P1L24 ... for semidiurnal component and so on are ...

*Corrected.*

P3L2 trapped tidal mode in SABER? in MERRA?

*In Sakazaki et al., 2018, it is mentioned that a representation of a trapped tidal mode would lead to differences between SABER and MERRA tidal amplitudes. The reason would be either an underestimation of the stratospheric ozone heating in the forecast model of MERRA or some systematic local time biases in retrieving the SABER temperatures. We have added this information in the manuscript (P3L2-3).*

P3L3 please reformulate follow similiar as GWs poloarization relations?

*We have rephrased this.*

P3L13 define s in k and motivate why is there a suffix s at the scale height

*This was just a typo. We have corrected this. The horizontal wave vector k is related to the zonal wave number s in P3L13.*

P3L19 Probably. The only one which I could think of is CORAL at Rio Grande. (just check, no action required, if still holds)

*The CORAL system is unfortunately still working only during the night, so it is not possible to conduct a multi-day time-series without a gap during the daylight conditions.*

P3L28 as well as fluctuations/variations due to GWs

*Corrected.*

P5L5 ... but ?here?in that work? without ...

*The work by Stober et al., 2017 was done without using a vertical information. We have slightly changed the sentence to make this clear.*

P5L14 variety -> variance

*Changed.*

P5L19 somehow the assumption and the regularization seem to be described twice -choose the more precise formulation

*We have reformulated this.*

P5L23 Studies such as ...

*Corrected.*

P5L27 if you filter them out that could be a self-fulfilling - but I think you are right

*Just from the observations without applying any filter the vertical wavelengths as well as the periods are quite large, therefore it is likely that the first Hough modes are the dominating tidal contribution.*

P5L29 It appears -> It turns out ??

*We rephrased this.*

P6L7 result section

*Corrected.*

P6L10 time resolved *what*

*Corrected.*

P6L29 For the left column this could be either coarser vertical resolution or MERRA's horizontal resolution which is filtering in a different dimension than the ones shown/discussed here.

*We agree that this could also be due to a coarser vertical resolution of MERRA-2. We have added this in the manuscript (P7L6).*

P7L1 MERRA data are snap-shot data? Then only the temporal sampling is coarser, but the temporal resolution should be sufficient.

*The temporal resolution of MERRA-2 data is 3h, while the lidar data is only integrated over 2h, so this is slightly better for the lidar. The differences for the vertical resolution are larger between both data sets, especially for the upper altitude levels, because for MERRA-2 the resolution is decreasing while this is constant at 1km for the lidar.*

P7l6 damps

*Corrected.*

P8l6 as far as it makes sense to compare the values of different units - this becomes a meaningful result, if you set it in relation to the expected relations for the maxima of the Hough modes

*We thank the reviewer for this hint. From Lindzen, 1967, Quart. J. Roy. Meteorol. Soc, it can be seen that the response presented for the location of Kuehlungsborn is fairly in agreement to the model results for the Hough functions of the diurnal component. We have included this information in the manuscript (P8L15-16).*

P9L21 Why should the amplitude influence the phase? Is there a second mode competing? Are you just speculating about a higher uncertainty? -> P10L1 o.k., but you have never discussed limits so far.

*The statement given in the manuscript refers only to the determination of the phase using the harmonic fit. This is just a technical comment regarding the uncertainty of the fit. If the*

*amplitude for one particular tidal component is, e.g., below 0.2 K for the temperature data, the signal is too insignificant for a proper phase determination.*

P10L3 My interpretation of the phase jump would have been that the phase cyclically continues in your fitting interval (the phase jumps are 12h = 360∘). Whether the point around 12 UT is a real independent point may depend on your fitting technique. Would your technique allow the phase to continuously increase even beyond 360∘? O.k. the fact that the phase shifts with respect to the diurnal tide makes the semidiurnal tide visible in F4. The main point should be still the relative amplitude ratio.

*The interpretation of a cyclical phase behavior is also possible in this case as the phase has an uncertainty of 360°. But however, the main point here is the phase difference between the single tidal components, which is not affected if each semidiurnal component is shifted in the same way. The fitting procedure is designed in such a way that phase jumps are considered and do not affect the analysis.*

F7-F9: Why are you using contour lines for the phases? That makes the plots very hard to read.

*From our point of view, the figures are even worse if we would use the same colormap as for the amplitudes. A stable phase could be seen as a horizontal straight line at a certain altitude. This is much more challenging to be inferred from a classical contour map plot. There is can be hard to see small changes in the color due to small drifts of the phase.*

P16L18 in very good agreement

*Corrected.*